# *Mastomys natalensis* Has a Cellular Immune Response Profile Distinct from Laboratory Mice

**DOI:** 10.3390/v13050729

**Published:** 2021-04-22

**Authors:** Tsing-Lee Tang-Huau, Kyle Rosenke, Kimberly Meade-White, Aaron Carmody, Brian J. Smith, Catharine M. Bosio, Michael A. Jarvis, Heinz Feldmann

**Affiliations:** 1Laboratory of Virology, Division of Intramural Research, National Institute of Allergy and Infectious Diseases, Rocky Mountain Laboratories, National Institute of Health, Hamilton, MT 59840, USA; kyle.rosenke@nih.gov (K.R.); kmeade-white@niaid.nih.gov (K.M.-W.); 2Research Technologies Branch, Rocky Mountain Laboratories, NIAID, NIH, Hamilton, MT 59840, USA; acarmody@niaid.nih.gov; 3Rocky Mountain Veterinary Branch Division of Intramural Research, National Institute of Allergy and Infectious Diseases, Rocky Mountain Laboratories, National Institute of Health, Hamilton, MT 59840, USA; brian.smith2@nih.gov; 4Laboratory of Bacteriology, Division of Intramural Research, National Institute of Allergy and Infectious Diseases, Rocky Mountain Laboratories, National Institute of Health, Hamilton, MT 59840, USA; bosioc@niaid.nih.gov; 5Faculty of Health: Medicine, Dentistry and Human Sciences, School of Biomedical Sciences, University of Plymouth, PL4 8AA, UK; michael.jarvis@plymouth.ac.uk; 6The Vaccine Group (TVG) Ltd., 14 Research Way, Derriford Research Facility, Plymouth Science Park, Plymouth PL6 8BU, UK

**Keywords:** *Mastomys natalensis*, immune response, T cell, effector cytokines, concanavalin A, phytohaemagglutinin P, lipopolysaccharide, Lassa virus

## Abstract

The multimammate mouse (*Mastomys natalensis; M. natalensis*) has been identified as a major reservoir for multiple human pathogens including Lassa virus (LASV), *Leishmania* spp., *Yersinia* spp., and *Borrelia* spp. Although *M. natalensis* are related to well-characterized mouse and rat species commonly used in laboratory models, there is an absence of established assays and reagents to study the host immune responses of *M. natalensis*. As a result, there are major limitations to our understanding of immunopathology and mechanisms of immunological pathogen control in this increasingly important rodent species. In the current study, a large panel of commercially available rodent reagents were screened to identify their cross-reactivity with *M. natalensis.* Using these reagents, ex vivo assays were established and optimized to evaluate lymphocyte proliferation and cytokine production by *M. natalensis* lymphocytes. In contrast to C57BL/6J mice, lymphocytes from *M. natalensis* were relatively non-responsive to common stimuli such as phytohaemagglutinin P and lipopolysaccharide. However, they readily responded to concanavalin A stimulation as indicated by proliferation and cytokine production. In summary, we describe lymphoproliferative and cytokine assays demonstrating that the cellular immune responses in *M. natalensis* to commonly used mitogens differ from a laboratory-bred mouse strain.

## 1. Introduction

*Mastomys natalensis*, a member of the *Muridae* family [1], has high prevalence across sub-Saharan Africa [2,3,4]. *M. natalensis* frequently lives in close association with humans and has been identified as a host reservoir for several zoonotic pathogens, including LASV [5,6,7], *Leishmania major* (*L. major*) [8], *Borrelia* spp. [9,10,11], and *Yersinia pestis* [12]. In contrast to humans, infection of *M. natalensis* by many of these zoonotic pathogens appears to be asymptomatic. How the immune system plays a role in pathogen persistence and clearance in these animals is unknown. An improved understanding of mechanisms by which *M. natalensis* controls microbial infection and transmission may lead to the development of novel intervention strategies to reduce zoonotic transmission to humans. 

Laboratory mouse and rat models have provided invaluable insight into the pathology and immunobiology of many different pathogens [13,14,15,16,17,18]. However, it is becoming increasingly appreciated that many aspects of microbial immunobiology may differ in these established rodent models from those of wild rodent species serving as pathogen reservoirs. Further, the applicability of commercially available reagents and well-established immunological techniques, including flow cytometry and in vitro T-cell assays, commonly used to study immune responses in laboratory mice and rats have not been established for the study of *M. natalensis* immunity.

CD4^+^ and CD8^+^ cytotoxic T cells play a crucial role in many antimicrobial immune responses via the production of effector cytokines, such as interferon gamma (IFN-γ) and tumor necrosis factor alpha (TNF-ɑ), leading to eradication and protective immunity against a range of microbial pathogens. Upon activation, naïve CD4^+^ T cells differentiate into distinct T cell subsets (Th1, Th2, Th17, Tfh, Treg) based on signals from the antigenic environment and interactions with antigen-presenting cells (APCs) [19]. In response to viral [20,21,22], parasitic [23,24,25,26], or bacterial [27,28] infections, CD4^+^ T cells predominantly differentiate into Th1 cells that produce inflammatory cytokines (i.e., IFN-γ and TNF-ɑ) and participate in cell-mediated immune responses, such as enhancement of the differentiation of naïve CD8^+^ T cells into cytotoxic T cells (CTL) for the clearance of infections of viral [20,21,29], bacterial [30,31,32], and parasitic [33,34] origins. CD4^+^ T cells can also mediate B cell differentiation and antibody production against extracellular [35,36,37] and intracellular [38,39] pathogens.

In the present study, we first aimed to determine if conventional reagents used in laboratory rodent studies could be used to trigger *M. natalensis* cells. We screened commercially available antibodies for use with *M. natalensis* splenic lymphocytes. Using identified reagents, we optimized in vitro assays for T-cell proliferation and the detection of IFN-γ and TNF-ɑ production. We show that in response to well-defined stimuli, the activation potential of *M. natalensis* splenic lymphocytes differs substantially from those observed in C57BL/6J mice.

## 2. Materials and Methods

### 2.1. Animals

In this study, we used *M. natalensis* from an in-house breeding colony originally established from rodents captured in Doneguebougou, Mali [40]. C57BL/6J mice were obtained from Jackson Laboratory. For all experiments, we used 5–7 week old animals with equal sex distribution. The number of animals used for each experiment is indicated in the legends of the figures. *M. natalensis* were free of ectromelia virus, mouse rotavirus, lymphocytic choriomeningitis virus, mouse adenovirus, Sendai virus, mouse hepatitis virus, minute mouse virus, mouse parvovirus, mouse polyoma virus, mouse norovirus, Theiler’s murine encephalomyelitis virus, *Mycoplasma pulmonis*, pinworms, and ectoparasites according to dirty bedding serology and filter EDx PCR testing (IDEXX BioAnalytics, Columbia, MO, USA). C57BL/6J mice were free of the above pathogens according to vendor reports. Animal studies were approved by the Institutional Animal Care and Use Committee and were conducted in compliance with all institutional and national guidelines for use and handling of animals.

### 2.2. Reagents and Antibodies

A list of reagents and antibodies is provided in Table 1.

### 2.3. Tissue Preparation

Spleens were harvested and placed in RPMI supplemented with 10% FBS and 2% of penicillin/streptomycin (ThermoFisher, Carlsbad, CA, USA), L-glutamine (Sigma, St. Louis, MO, USA), and 0.5 mM of β-2-mercaptoethanol (Sigma, St. Louis, MO, USA) (cRPMI). Tissues were individually dissociated at room temperature (RT) in Miltenyi dissociator C tubes (Miltenyi Biotec, San Diego, CA, USA) (Table 1). Following dissociation, spleen homogenates were passed through a cell strainer (Fisher Scientific, Pennsylvania, PA, USA) and centrifuged at 700× *g* for 5 min. Red blood cell were lysed using 1× RBC Lysis Buffer according to manufacturer’s instructions (eBioscience™ ThermoFisher, Carlsbad, CA, USA). Remaining cells were washed and resuspended in cRPMI, and cell counts were performed by mixing 10 μL of sample with 10 μL of 0.4% trypan blue solution. The mixture was loaded onto a chamber slide and counted using a Countess cell counter (Bio-Rad, Hercules, CA, USA).

### 2.4. T Cell Proliferation

To analyze T cell proliferation, splenocytes were stained with CellTrace Violet (CTV, ThermoFisher, Carlsbad, CA, USA) prior to stimulation with mitogens. Splenocytes (5 × 10^5^ cells/96-well round bottom plate) were added to triplicate wells and stimulated with three different mitogens: concanavalin A (ConA; 1× ThermoFisher, Carlsbad, CA, USA), phytohaemagglutinin P (PHA; 1.5%, ThermoFisher, Carlsbad, CA, USA), and lipopolysaccharide (LPS; 10 μg/mL, Sigma, St. Louis, MO, USA). Mitogen stimulation was performed either in the presence or absence of mouse interleukin (IL)-2 (25 IU/mL; Miltenyi, San Diego, CA, USA) for up to 6 days in cRPMI at 37 °C, 5% CO_2_ (Table 1).

### 2.5. Extracellular Staining for Flow Cytometry

Non-specific binding was blocked using TruStain (Biolegend, San Diego, CA, USA) for 10 min. Splenocytes were stained with T cell surface markers with rat anti-CD3, CD8, and CD4 for 30 min at 4 °C (Table 1). The samples were analyzed on a BD FACS Symphony instrument (BD Biosciences, San Jose, CA, USA) and analyzed by FlowJo v10.

### 2.6. Intracellular Staining for Flow Cytometry

T cells were re-stimulated on day 6 with phorbol 12-myristate 13-acetate (PMA, 50 ng/mL, Sigma) and ionomycin (Iono; 1 μg/mL; Merck Calbiochem, Burlington, MA, USA) for 6 h in the presence of brefeldin A (BFA; 1×; eBioscience ThermoFisher, Carlsbad, CA, USA) at 37 °C, 5% CO_2_. Non-specific binding was blocked using TruStain (Biolegend, San Diego, CA, USA) for 10 min. Cell viability was assessed using live/dead eFluor780 (ThermoFisher, Carlsbad, CA, USA) for 20 min at 4 °C (Table 1). Then the cells were fixed and permeabilized (Intracellular Fixation & Permeabilization Buffer Set; ThermoFisher, Carlsbad, CA, USA) and stained for rat anti-CD3, a cytoplasmic epitope of CD3 (Bio-Rad, Hercules, CA, USA) and selected intracellular proteins (anti-mouse TNF-α; anti-rat IFN-γ) (Table 1), for 45min at RT in permeabilization wash buffer (eBioscience ThermoFisher, Carlsbad, CA, USA). The samples were analyzed using the BD FACS Symphony instrument (BD Biosciences, San Jose, CA, USA) and FlowJo v10.

### 2.7. Cytometric Bead Array (CBA)

Splenocytes (1 × 10^6^ cells/well) were incubated with or without ConA, LPS, or PHA mitogens for 24 h in cRPMI as technical duplicate replicates. Supernatants were collected and stored at −20 °C until use. TNF-α and IFN-γ release into the supernatant was measured by CBA using anti-mouse TNF-α and anti-rat IFN-γ antibodies according to the manufacturer’s instructions (Table 2).

### 2.8. Software and Statistical Analysis

Flow cytometry data were analyzed using FlowJo software v10 (Tree Star). Statistical analyses were performed using the Prism software v8 (GraphPad, San Diego, CA, USA). Wilcoxon non-parametric test and one-way ANOVA were used. Variance was similar between the groups being compared.

## 3. Results

### 3.1. Commercial Rat and Mouse Antibodies Cross-React with M. natalensis T Cell Receptors and Intracellular Cytokines

Commercial rat and mouse antibodies against T cell receptors (CD3, CD8, CD4) and effector molecules (TNF-α and IFN-γ) were evaluated for their cross-reactivity with *M. natalensis* splenocytes (Table 3). Spleens from *M. natalensis* were harvested and stained with T cell receptor antibodies from different clones and analyzed by flow cytometry. We found that *M. natalensis* CD3 and CD8b receptors were recognized by rat anti-CD3 *clone CD3-12* and rat anti-CD8b *clone 341*, respectively. No CD4 antibodies tested in this study cross-reacted with *M. natalensis* splenocytes (Table 3). In addition, we could demonstrate that *M. natalensis* IFN-γ and TNF-α cytokines were recognized by rat anti- IFN-γ, clone DB-1, and mouse anti- TNF-α *clone MP6-XT22,* respectively (Table 3).

### 3.2. ConA Mitogen Efficiently Induced M. natalensis T Cell Proliferation In Vitro

ConA [41,42], PHA [43,44,45], and LPS [46,47] are the most commonly used mitogens targeting lymphocytes as they do not require antigen presentation to activate T cells and have been used to describe general immune responses, such as proliferation and cytokine production, in these cell populations. To optimize the in vitro assay for the induction of T cell proliferation and differentiation into mature effector cells, spleens from *M. natalensis* were harvested and stimulated with these three mitogens and T cell proliferation was measured at different time points (0, 3, 4, 5, and 6 days) post-stimulation. Recombinant mouse IL-2 was added to a subset of these samples to assess the impact of IL-2 signaling on activated cell survival, which by itself failed to stimulate T-cell proliferation. The CTV-based assay has been used to quantify T-cell proliferation in response to different mitogens. Unstimulated splenocytes were used as negative controls.

In agreement with previous studies, LPS, ConA, and PHA mitogens induced proliferation among C57BL/6J splenic lymphocytes [42,43,47] independently of IL-2 (Figure 1a–c). In contrast to C57BL/6J, *M. natalensis* splenic lymphocytes stimulated with LPS did not proliferate with or without IL-2 (Figure 1a). We observed that *M. natalensis* cells stimulated with PHA proliferate only in the presence of recombinant mouse IL-2 (Figure 1b). Finally, ConA was sufficient to enhance a strong CD3^+^-T cell proliferation in both C57BL/6J and *M. natalensis*, and this effect was independent of IL-2 (Figure 1c). Thus, ConA triggered the most efficient proliferation of *M. natalensis* CD3^+^ T cells compared to PHA and LPS mitogens.

### 3.3. Comparative Secretion of Effector Molecules in M. natalensis in Response to Stimuli

To further examine the differential immune response profiles by *M. natalensis* splenic lymphocytes, we assessed the level of cytokines in cell supernatant or secreting cells following mitogenic stimulation. *M. natalensis* splenic lymphocytes were stimulated with LPS, PHA, or ConA for 24 h. C57BL/6J splenic lymphocytes were used as a positive control. As described above, the presence of IL-2 can have an impact on T cell proliferation (Figure 1b); therefore, we also included recombinant mouse IL-2 to a subset of samples to determine its impact on cytokine production. TNF-α and IFN-γ were detectable in the supernatant of secreting splenic lymphocytes from C57BL/6J mice when stimulated with mitogens as determined by mouse specific ELISA, CBA, and ELISpot kits. No detectable TNF-α and IFN-γ in cell culture supernatants from *M. natalensis* splenic lymphocytes was observed when assessed by ELISA (data not shown). However, both cytokines were detected in supernatant utilizing CBA (Table 2). Further, TNF-α was readily detected by ELISpot (Table 2).

We used CBA specific to rat or mouse to measure cytokine responses to different stimuli in *M. natalensis* and C57BL/6J rodents, respectively. C57BL/6J splenic lymphocytes responded to all stimuli, LPS and ConA, as indicated by the production of both TNF-α and IFN-γ, and no differences were observed in IL-2 treated groups. Among PHA-stimulated C57BL/6J cells, the production of both cytokines significantly decreased (*p* < 0.01) in the presence of IL-2 (Figure 2a). In contrast, IL-2 did not impact C57BL/6J T-cell proliferation (Figure 1a–c) and *M. natalensis* stimulated cells only produce TNF-α in response to PHA but not IFN-γ independent of the presence of IL-2 (Figure 2b). However, LPS and ConA treatment significantly increased the secretion of TNF-α and IFN-γ within 24 h by *M. natalensis* splenic lymphocyte. Addition of IL-2 did not significantly change the amount of either cytokine produced under these conditions (Figure 2a,c). Taken together, LPS and ConA efficiently induce both IFN-γ and TNF-α secretion independently of IL-2 in both C57BL/6J and *M. natalensis*.

### 3.4. Comparative Expression of Intracellular Cytokines in M. natalensis in Response to Different Stimuli

Next, we assessed the expression of effector molecules by *M. natalensis* CD3^+^ T cells in response to mitogens stimuli. Intracellular IFN-γ and TNF-α were not detected among *M. natalensis* splenic lymphocytes stimulated only with PMA/Iono in the presence of BFA for 6 h. Therefore, *M. natalensis* splenocytes were stimulated with ConA, LPS, or PHA for 6 days followed by re-stimulation with PMA/Iono in the presence of BFA (Figure 3b). All mitogens induced the expression of both IFN-γ and TNF-α among positive control C57BL/6J CD3^+^ T cells (Figure 3b,d). All three mitogens also induced the expression of TNF-α by *M. natalensis* CD3^+^ T cells, but only ConA induced significant expression of both IFN-γ and TNF-α (Figure 3b,c). Notably, as observed in studies assessing the proliferation potential, IL-2 did not impact the expression of either IFN-γ or TNF-α in the *M. natalensis* CD3^+^ cell population. Therefore, only ConA efficiently induced the expression of IFN-γ and TNF-α by *M. natalensis* CD3^+^ T cells as detected by intracellular cytokine staining.

## 4. Discussion

*M. natalensis* is a host for multiple emerging and re-emerging human pathogens (i.e., LASV, *Leishmania* spp., *Yersinia* spp., and *Borrelia* spp.). It is unknown how these rodents survive infection with these pathogens to serve as vectors for transmission to humans. Understanding this paradigm may ultimately help the development of new therapeutic strategies. T cells play an important role in the defense against microorganisms, in part by secreting key mediators which enable eradication of the infecting agent. Th1 T cells are characterized by their property to produce IFN-γ, TNF-α, and IL-2. These cells play a central role in mediating adaptive immune responses to microbial agents [21,23,32,33,35,48,49], tumor [50], inflammation, and autoimmune diseases [51,52]. Therefore, development of assays that assess these responses in *M. natalensis*-derived cells would provide valuable insight into our understanding of ongoing immune responses in this reservoir host.

To study the T cell-mediated immunity in *M. natalensis*, we identified, optimized, and established immunological techniques used for laboratory mice and rats to trigger T cell activation. *M. natalensis* splenocytes were stimulated with the classical mitogens (LPS, PHA, or ConA) that do not rely on antigen specificity or presentation to trigger proliferation and production of cytokines by these cells [41,42,43,44,45,46,47]. We found that *M. natalensis* CD3^+^, CD8^+^ T cell markers, and IFN-γ were largely recognized by antibodies directed against rat proteins whereas TNF-α was detected by antibodies targeting mouse cytokines (Table 3). Approaches for measuring cytokines/chemokines associated with cytotoxic T-lymphocyte function in response to mitogenic stimuli in cell supernatant or cytokine-secreting cells were also assessed. We found that only CBA, a bead-based immunoassay, was capable of measuring both *M. natalensis* IFN-γ and TNF-α in cell culture supernatants after 24 h of stimulation with mitogens (Figure 2). CBA, ELISA, and ELISpot assays all use primary (capture) and secondary (detection) antibodies. However, there are differences among these methods, such as their sensitivity to detect low frequency of cytokine-secreting cells (ELISpot) or cytokines release into cell culture (CBA and ELISA) [53,54,55]. In addition, we have demonstrated that *M. natalensis* IFN-γ is only detectable by a rat-IFN-γ CBA kit and TNF-α by mouse- TNF-α ELISpot and CBA kits. These results suggest that paired antibodies used to detect IFN-γ or TNF-α may differ in those assays or that some antibodies do not function due to lack of binding. This is supported by a recent phylogenetic study demonstrating that the genome of *Mastomys coucha* aligns to 90.1% with mouse and 85.5% with rat, supporting an intermediate position of *M. natalensis* in the rodent taxonomy [56].

In this study, we have demonstrated significant differences in T cell proliferation and cytokine production between *M. natalensis* and C57BL/6J splenic lymphocytes in response to different stimuli. It should be noted, however, that *M. natalensis* animals from our colony were recently derived from wild caught animals [40] and thus may harbor microorganisms distinct from laboratory C57BL/6J mice derived from a clean and defined laboratory environment. Therefore, *M. natalensis* immune responses to experimental mitogen stimulation may be reduced due to continuing exposure to microbial stimuli [57,58].While all mitogens induced C57BL/6J splenic lymphocytes proliferation, independent of IL-2, LPS was not effective at triggering proliferation of *M. natalensis* splenic T cells regardless of whether exogenous IL-2 was present or not (Figure 1a). Numerous studies using mouse and human cells have investigated the effect of LPS on cell proliferation [59] and cytokine production by lymphocytes [60]. LPS has been noted to both activate [61] and inhibit [62] lymphocytic activation. However, most of the studies have shown that the mechanism of T cell activation by LPS is mediated by innate cells, such as monocytes or APCs, providing costimulatory molecules signals via direct cell contact [61,63]. Therefore, our results suggested that LPS may not trigger appropriate responses by *M. natalensis* APCs that support T cell proliferation in our model. However, the lack of specific antibodies and immunology tools to *M. natalensis* currently does not allow us to confirm the role of LPS on *M. natalensis* APCs or T cells. In contrast, *M. natalensis* splenic lymphocytes stimulated with LPS for 24 h increased secretion of TNF-α and IFN-γ which was also independent of IL-2 (Figure 2a). This may be explained in part by differences of transcription, translation, protein processing, export, and protein degradation by each species of rodent [64]. Guy et al. also demonstrated that the large number of immunoreceptor tyrosine activation motifs (ITAM) within the T cell receptor (TCR)-CD3 complex (TCR-CD3 ITAM) play an important role in T cell development and function. Indeed, they have shown that low CD3 ITAM engage TCR-driven pathways that lead to cytokine production while high TCR-CD3 ITAM multiplicity promote T cell proliferation. These results support that proliferation and cytokine production can be two distinct events in T cells, dependent on the TCR signaling [65].

We also demonstrated that *M. natalensis* splenocytes stimulated with PHA require the presence of IL-2 to proliferate, while PHA efficiently induce C57BL/6J splenic lymphocytes proliferation without IL-2 cytokine (Figure 1b). Regardless of the presence/absence of IL-2, PHA was efficient to induce secretion of TNF-α but not IFN-γ by *M. natalensis* CD3^+^ T cells. However, IL-2 significantly decreased the production of both cytokines by C57BL/6J cells (Figure 2b). The decrease of IFN-γ and TNF-α secretion by C57BL/6J splenic lymphocytes stimulated with PHA in the presence of IL-2 suggests that exogenously added IL-2 may have induced T cell exhaustion resulting in functional impairment of T cells to secrete IFN-γ and TNF-α. Further studies need to confirm this hypothesis. We hypothesize that PHA combined with IL-2 may induce IL-2 receptor expression by T cells [45], resulting in enhanced proliferation by *M. natalensis* cells. However, lack of specific antibodies against *M. natalensis* do not allow us to confirm the effect of PHA on IL-2 receptor at this time. Finally, we demonstrated that ConA was sufficient to stimulate *M. natalensis* T cell proliferation and differentiation into effector T cells in the absence of IL-2 (Figure 1 and Figure 3) and significantly increased the secretion of TNF-α and IFN-γ from *M. natalensis* splenic lymphocytes independent of the presence or absence of IL-2 (Figure 2c). This suggests that ConA by itself triggered cross-linking of the TCR complex which leads to T cell proliferation and cytokine secretion among *M. natalensis* splenic cells, contrary to LPS and PHA mitogens [42]. Still, the molecular mechanisms by which TNF-α and IFN-γ genes expression and secretion occurs in response to each mitogen remain to be elucidated.

## 5. Conclusions

To conclude, only ConA was a strong stimulator of proliferation and differentiation into effector cells of *M. natalensis* CD3^+^ T cells. Thus, ConA-stimulating assays will allow us to determine ranges for IFN-γ and TNF-α in response to mitogen stimulation for *M. natalensis*-derived cells by flow cytometry (intracellular staining and CBA). These assays will be used to characterize the immune response in *M. natalensis* against infection and we believe this understanding of differences in distinct immune responses provides a critical underpinning for future studies on the immune response to pathogen infection in an increasingly important reservoir species. This is an important first step in the development of assays designed to understand the role of the innate and adaptive immune responses in this important reservoir species. Moving forward, there is a need to identify and optimize more immunology tools for *M. natalensis*, such as assays assessing cytotoxicity of CD4/CD8 T cells in vivo and in vitro.

## Figures and Tables

**Figure 1 viruses-13-00729-f001:**
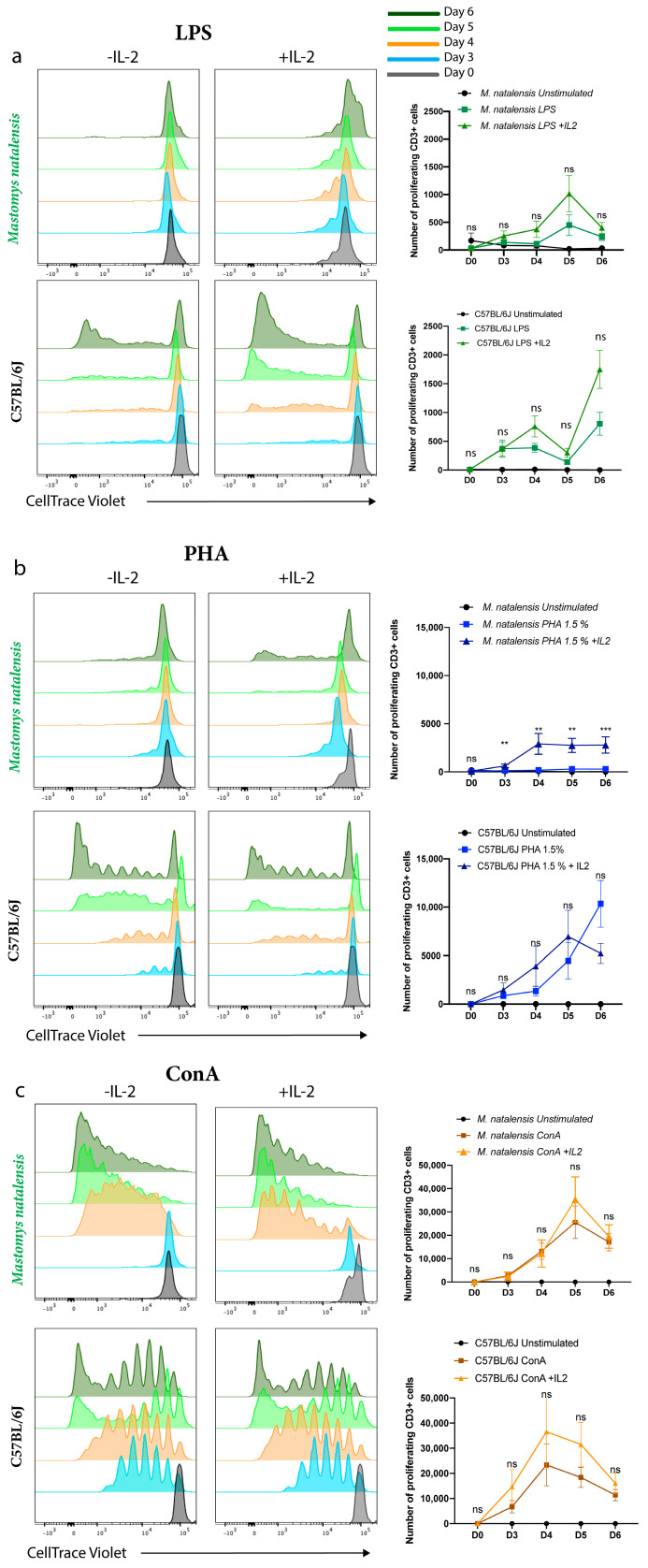
T cell proliferation in response to different stimuli. Splenic cells derived from *M. natalensis* (N = 12) or C57BL/6J (N = 12) were stimulated with LPS (**a**), PHA (**b**), and ConA (**c**) mitogens in the presence or absence of IL-2 cytokines. A CTV-based assay has been used to quantify T-cell proliferation at different time points. Number of proliferating CD3^+^ T cells of *M. natalensis* and C57BL/6J are shown in the upper and bottom graph of parts a, b, and c, respectively. Mean ± SEM of three independent experiments. ns: not significant; ** *p* < 0.01; *** *p* < 0.001; Wilcoxon non-parametric test.

**Figure 2 viruses-13-00729-f002:**
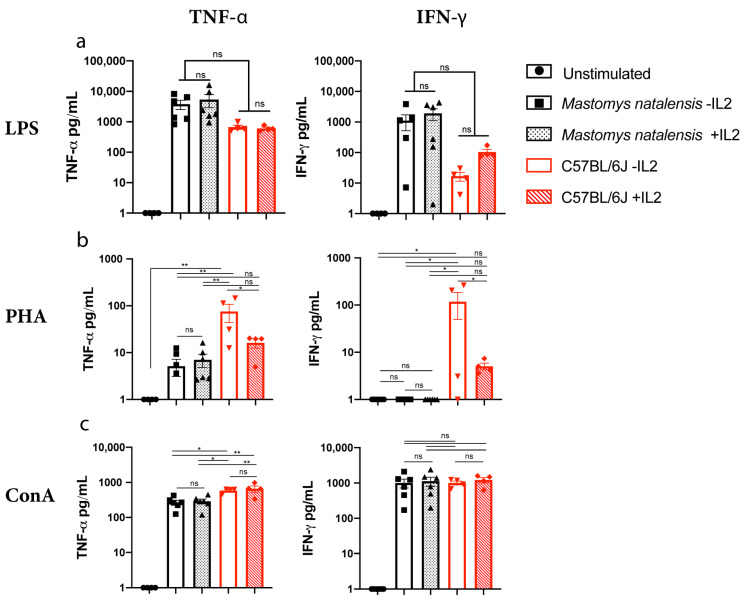
Secretion of effector molecules in response to stimuli. Splenic cells derived from *M. natalensis* (N = 6) or C57BL/6J (N = 4). Mice were stimulated with LPS (**a**), PHA (**b**), or ConA (**c**) for 24 h. Secretion of IFN-γ and TNF-α (**a**–**c**) were measured in the supernatants by CBA. Symbols represent individual animals. Mean ± SEM is shown. ns: not significant; * *p* < 0.01; ** *p* < 0.001; One-way ANOVA.

**Figure 3 viruses-13-00729-f003:**
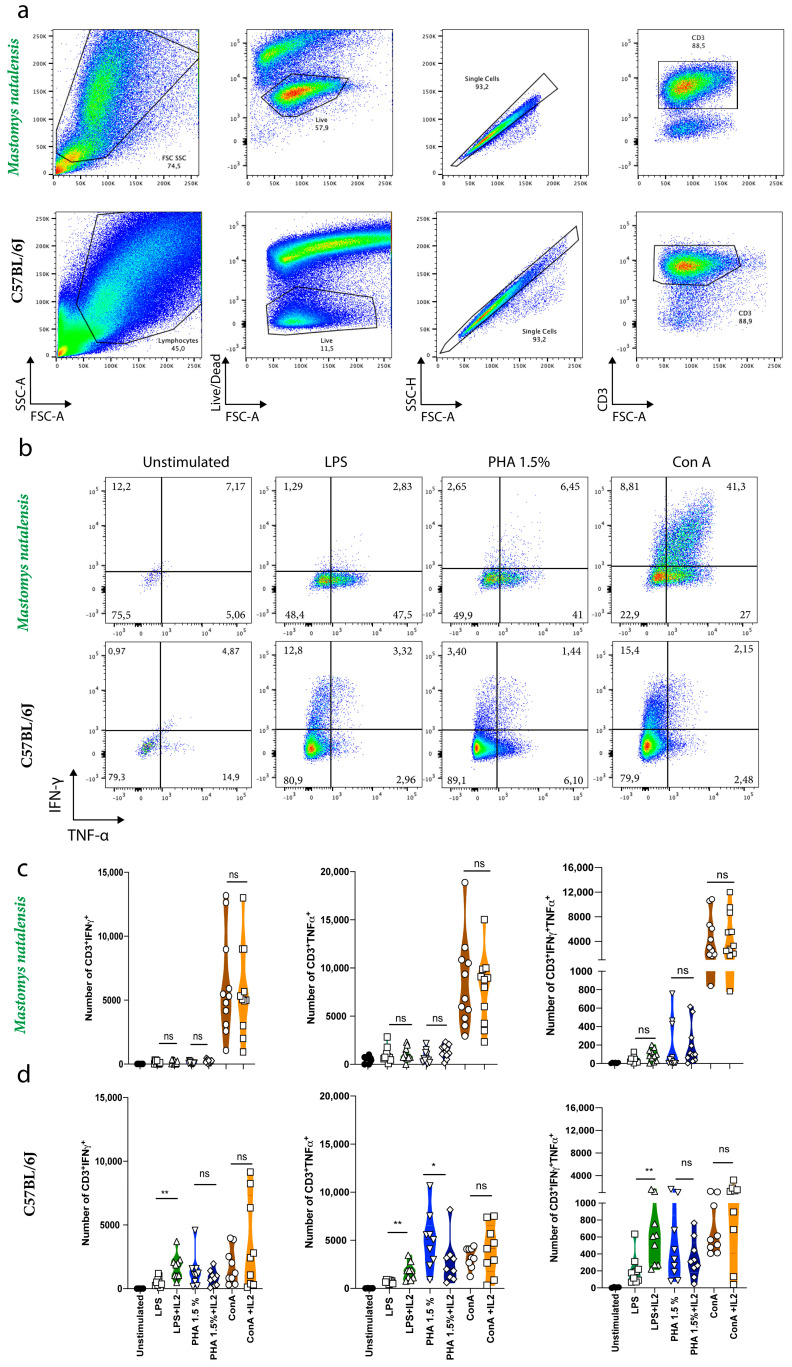
Expression of intracellular cytokines in response to different stimuli. Exemplary gating strategies defining the investigated CD3^+^ T cell population of *M. natalensis* and C57BL/6J are shown in the upper and bottom portion of graph a, respectively (**a**); Representative flow cytometry plots of splenic cells derived from *M. natalensis* (upper part) or C57BL/6J mice (bottom part, used as a positive control) stimulated with LPS, PHA, or ConA for 6 days followed by 6h stimulation with PMA/Ionomycin and BFA (**b**); Expression of TNF-α and IFN-γ were assessed by intracellular staining (**b**); Number of CD3^+^ T cells expressing effector molecules by *M. natalensis* (**c**); and C57BL/6J (**d**) is shown. Symbols represent individual animal. N = 12 *M. natalensis* and N = 9 C57BL/6J mice of two independent experiments (**c**,**d**). ns: not significant; * *p* < 0.01; ** *p* < 0.001; One-way ANOVA.

**Table 1 viruses-13-00729-t001:** List of reagents and antibodies used in this study.

**Mechanical and Tissue Dissociation**		
**Reagents**	**References**	**Vendors**
Gibco Fetal Bovine Serum	16000044	ThermoFisher
RPMI	R8758	Sigma-Aldrich
Pencillin/Streptomycin	15070063	ThermoFisher
L-glutamine	25030164	ThermoFisher
β-Mercaptoethanol	M3148-25ML	Sigma-Aldrich
Miltenyi dissociator C tubes	130-096-334	Miltenyi
Cell strainer (40 μm)	22-363-547	Fisherscientific
Red blood cell lysis (1× RBC Lysis Buffer)	00-4333-57	ThermoFisher
**In vitro Lymphocytes stimulation and proliferation**		
**Reagents**	**References**	**Vendors**
Concanavalin A	00-4978-03	ThermoFisher
Phytohaemagglutinin P	10576015	ThermoFisher
Lipopolysaccharide	L2630-10MG	Sigma-Aldrich
Mouse interleukin (IL)-2	130-120-331	Miltenyi
CellTrace Violet	C34557	ThermoFisher
**Reagents and antibodies for flow cytometry**		
**Reagents**	**References**	**Vendors**
Phorbol 12-myristate 13-acetate	P8139-1MG	Sigma-Aldrich
Ionomycin	407950-1MG	Merck Calbiochem
Brefeldin A	00-4506-51	ThermoFisher
TruStain FcX	101320	BioLegend
Fixable Viability Dye eFluor™ 780	65-0865-14	ThermoFisher
Intracellular Fixation & Permeabilization Buffer Set	88-8824-00	ThermoFisher
Rat anti Human CD3 FITC	MCA1477F	Bio-Rad
Anti-mouse TNF-α Brilliant Violet 785	506341	BioLegend
Mouse Anti-Rat IFN-γ PE	559499	BDbiosciences

**Table 2 viruses-13-00729-t002:** Commercial kits used for the detection of cytokines produced by Mastomys-derived splenic lymphocytes.

	Reagents	Source	Catalog Number	Cross-React with Mastomys
**Cytometric Bead Array (CBA)**	**Rat IFN-γ Flex** **Set**	**BD**	**558305**	**Yes**
	**Mouse TNF-α Flex Set**	**BD**	**558299**	**Yes**
	**Mouse/Rat** **Soluble Protein Master Buffer Kit**	**BD**	**558266**	**Yes**
**ELISPOT**	Rat IFN-γ Single color	ImmunoSpot		No
	**Mouse TNF-α Single color**	**ImmunoSpot**		**Yes**
**ELISA**	Mouse TNF-α ELISA MAX™Standard Set	BioLegend	430901	No
	Mouse IFN-γ ELISA MAX™Standard Set	BioLegend	430801	No
	Purified Rat Anti-Mouse IFN-γ	BD	551309	No
	Biotin Anti-Mouse IFN-γ	BD	551506	No
	Biotin Rat Anti-Mouse IFN-γ	BD	554410	No
	Recombinant Rat IFN-γ	BD	550072	No

**Table 3 viruses-13-00729-t003:** Antibodies tested in this study.

Specificity	Antibody	Conjugate	Clone	Reference	Vendor	Cross-React with Mastomys
Mouse	Purified anti-mouse CD4	N/A	GK1.5	100401	Biolegend	No
	CD3e	PE	145-2C11	100307	Biolegend	No
	CD8a	APC	53.6.7	100711	Biolegend	No
	IFN-γ	APC	XMG1.2	505809	Biolegend	No
	**TNF-α**	**Brilliant Violet 785**	**MP6-XT22**	**506341**	**Biolegend**	**Yes**
	**TNF-α**	**PE**	**MP6-XT22**	**12-7321-41**	**eBioscience**	**Yes**
Rat/Human/Mouse	CD3	FITC	CD3-12	MCA1477F	Bio-Rad	Yes
Rat	CD3	FITC	G4.18	559975	BD	No
	**Purified anti-rat CD8b**	**N/A**	**341**	**200702**	**Biolegend**	**Yes**
	CD8b	PE	eBio341	12-0080-82	ThermoFisher	No
	CD8a	APC	G28	200609	Biolegend	No
	CD8a	BV421	OX-8	740041	BD	No
	CD8b	BV421	341	742915	BD	No
	Purified anti-rat CD8a	N/A	OX-8	201701	Biolegend	No
	Purified anti-rat CD4	N/A	W3/25	201501	Biolegend	No
	CD4	BV786	OX-35	740912	BD	No
	CD4	BV786	OX-38	743093	BD	No
	CD4	APC	W3/25	201509	Biolegend	No
	**IFN-γ**	**AF647**	**DB-1**	**562213**	**BD**	**Yes**
	**IFN-γ**	**PE**	**DB-1**	**559499**	**BD**	**Yes**
Rat/mouse/rabbit	TNF-α	PE	TN3-19.12	559503	BD	No

Note: antibodies that recognized *M. natalensis* are highlighted in bold.

## Data Availability

The data presented in this study are available on request f, rom the corresponding author.

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
