# Peer review of "Mastomys natalensis Has a Cellular Immune Response Profile Distinct from Laboratory Mice"

_viruses, 2021, doi:10.3390/v13050729_

Round 1

Reviewer 1 Report

The manuscript entitled “Mastomys natalensis has a cellular immune response profile distinct from laboratory mice” is complete and well described.

In this manuscript, the authors investigated lymphocyte proliferation and cytokine response on M. natalensis by using the commercially available rodent reagents and their cross reactivity to them.  Their results confirm that there are significant differences in T cell proliferation and cytokine production between M. natalensis and C57BL/6J mice.

Overall, the data supports the conclusion, and the manuscript is well described.

Author Response

Dear reviewer,

You will find in attached our answers to your comments.

Best regards,

Tsing-Lee Tang-Huau

Reviewer 2 Report

The authors present results from a series of experiments to characterize reagents and methods to assess the cellular immune response in Mastomys natalensis as compared to a commonly used laboratory strain C57Bl/6J. While relatively few laboratories will be concerned with Mastomys natalensis the approach taken and the results should be interesting to readers who want to characterize the cellular immune response in other natural reservoirs. Still, a number of things should be addressed to improve the manuscript.

Abstract line 37-38; this sentence contains a number of duplication and should be rewritten for clarity. example ...the cellular immune response in M. natalensis to commonly used mitogens differ from a laboratory bred mouse strain.

The introduction Lines 61-71 describes the immune response to viruses alone but the potential importance of M. natalensis extends beyond that to bacterial and parasites. Please expand the introduction to include these pathogens. 

Line 72 determine "if" not "in"

M&M: Animals. Lines 81 - 84. First sentence needs to be rewritten for clarity. How many animals were used for each study? What was the distribution of males and females? Suggest replacing B6/J throughout document with the C57BL/6J or use the abbreviation in figure legends as well.

Tissue preparations: Were spleens pooled or processed individually?

Line 182: In "contrast" not "contrary"

Figure 1 legend needs to be more detailed. Which is from M. natalensis and which is from BL/6J? P values should be reported with a "." not a "," This applies to all figures.

Discussion Lines 314 - 319. How does this relate to the testing that was done in the M&M section.

Line 327 resulting "in" not "to"

Author Response

(The authors gave the same response as above.)

Reviewer 3 Report

In this manuscript the authors tested mouse and rat reagents to detect Mastomys natalensis T cell markers and cytokines. In addition, they standardized a series of immunological techniques to measure T cell responses in M. natalensis and compared with another mouse strain B/6J. Finding difference in T cell proliferation and differentiation between both mouse strains.

The following are some suggested changes to the manuscript:

Please check references 9 to 11. Reference 9 do not mention M. natalensis. Reference 10 do not mention Borrelia spp. You may consider using: Boardman K et al. Host Competency of the Multimammate Rat Mastomys natalensis Demonstrated by Prolonged Spirochetemias with the African Relapsing Fever Spirochete Borrelia crocidurae. 2019. Am J Trop Med Hyg.

Reference 11 refers to Lassa and Yersinia. You may consider to add Yersinia pestis to the text.

In table 3 please consider to highlight (e.g. bold letters) those antibodies that recognized M. natalensis markers.

In figures 2 and 3 please specify the mouse strain in the "Y" axes. I take it that boxes in the upper part are M. natalensis and in the bottom B/J6.  

Please check B6/J and B/6J in the text and used one of them.

In the sentence: "Finally, ConA was sufficient to enhance a strong CD3+-T cell proliferation and this effect was independent of IL-2". Please add: "in both B/6J and M. natalensis".

Why the ELISA and ELIspot data are not shown?

In the text you stated that : "M. natalensis stimulated cells did not produce IFN-γ or TNF-α in response to PHA" and refer to figure 2b. It looks like TNF-α is produce in about 5-10 pg/ml. Is this under the cut-off point to be considered positive? Please explain and change it in the text if neccesary.

In the sentence: "Taken together, LPS and ConA efficiently induce IFN-γ and TNF-α secretion independently of IL-2". Please add: "in both B/6J and M. natalensis".

In the discussion section, please organize references from 32 to 38.

Since LPS, PHA, and ConA are non-specific stimuli, I suggest to rephrase the  sentence: "To study the immune response to microbial infection in M. natalensis" as follow: "To study the T cell mediated immunity in M. natalensis".

Author Response

(The authors gave the same response as above.)
